# Matrix metalloproteinase 14 is required for fibrous tissue expansion

Susan H Taylor[1†], Ching-Yan Chloé Yeung[1†], Nicholas S Kalson[1†], Yinhui Lu[1], Paola Zigrino[2], Tobias Starborg[1], Stacey Warwood[1], David F Holmes[1], Elizabeth G Canty-Laird[3], Cornelia Mauch[2], Karl E Kadler[1*]

[1]Wellcome Trust Centre for Cell-Matrix Research, Faculty of Life Sciences, University of Manchester, Manchester, United Kingdom; [2]Department of Dermatology, Center for Molecular Medicine, University of Cologne, Cologne, Germany; [3]Department of Musculoskeletal Biology, Institute of Ageing and Chronic Disease, Faculty of Health and Life Sciences, University of Liverpool, Liverpool, United Kingdom

**Abstract** Type I collagen-containing fibrils are major structural components of the extracellular matrix of vertebrate tissues, especially tendon, but how they are formed is not fully understood. MMP14 is a potent pericellular collagenase that can cleave type I collagen in vitro. In this study, we show that tendon development is arrested in *Scleraxis-Cre::Mmp14* lox/lox mice that are unable to release collagen fibrils from plasma membrane fibripositors. In contrast to its role in collagen turnover in adult tissue, MMP14 promotes embryonic tissue formation by releasing collagen fibrils from the cell surface. Notably, the tendons grow to normal size and collagen fibril release from fibripositors occurs in *Col-r/r* mice that have a mutated collagen-I that is uncleavable by MMPs. Furthermore, fibronectin (not collagen-I) accumulates in the tendons of *Mmp14*-null mice. We propose a model for cell-regulated collagen fibril assembly during tendon development in which MMP14 cleaves a molecular bridge tethering collagen fibrils to the plasma membrane of fibripositors.

*For correspondence: karl.kadler@manchester.ac.uk

†These authors contributed equally to this work

Competing interests: The authors declare that no competing interests exist.

## Introduction

Matrix metalloproteinase 14 (MMP14, also known as membrane type I-MMP) is a member of the family of MMPs and contains a transmembrane domain for insertion into the plasma membrane (*Sato et al., 1994*). MMP14 has been implicated in cancer cell invasion (*Hotary et al., 2003*) and embryonic development (*Holmbeck et al., 1999*; *Zhou et al., 2000*) because of its ability to degrade extracellular matrix (ECM) macromolecules especially type I collagen (*Overall, 2001*; *Tam et al., 2002*; *Lee et al., 2006*; *Kessenbrock et al., 2010*; *Gialeli et al., 2011*). Mice deficient in MMP14 die within a few weeks of birth with generalized connective tissue abnormalities including osteopenia and soft tissue frailty (*Holmbeck et al., 1999*; *Zhou et al., 2000*). We were curious why absence of MMP14, which is an efficient collagenase in vitro, leads to connective tissue frailty rather than collagen accumulation.

Collagens are a large family of triple helical proteins that are widespread throughout the vertebrate body and are critical for tissue scaffolding (*Huxley-Jones et al., 2007*). More than 28 collagen and collagen-related proteins occur in vertebrate tissues of which type I collagen is the archetypal member of the subfamily of fibril-forming collagens (*Kadler et al., 2007*). The fibrils formed from type I collagen are the largest (with a mass per unit length up to ∼0.3 TDa/μm) and most size pleomorphic (from ∼1 μm to >1 mm) protein polymers in vertebrates and are essential for fibrous tissue development (*Schnieke et al., 1983*). Collagen fibril assembly has best been studied in embryonic tendon, which contains narrow diameter (∼50 nm) fibrils in which one end of

**eLife digest** A scaffold of proteins called the extracellular matrix surrounds each of the cells that make up our organs and tissues. This matrix, which contains fibres made of proteins called collagens, provides the physical support needed to hold organs and tissues together. This support is especially important in the tendons—a tough tissue that connects the muscle to bone—and other 'connective' tissues.

An enzyme called MMP14 is able to cut through chains of collagen proteins. It belongs to a family of proteins that are involved in breaking down the extracellular matrix to enable cells to divide and for other important processes in cells. Some cancer cells exploit MMP14 to enable them to leave their tissue of origin and spread around the body. Therefore, when researchers bred mutant mice that lacked MMP14, they expected to see excessive growth of collagen fibres in the connective tissues of the mice. However, these mice actually have extremely thin, fragile connective tissue and die soon after birth.

Earlier in 2015, a group of researchers demonstrated that the first stage of tendon development in mice involves the formation of collagen fibres, which are attached to structures that project from tendon cells called fibripositors. Then, soon after the mice are born, the fibripositors disappear and the collagen fibres are released into the extracellular matrix where they grow longer and become thicker. Now, Taylor, Yeung, Kalson et al.—including some of the researchers from the earlier work—have used electron microscopy to investigate how a lack of MMP14 leads to fragile tendons in young mice.

The experiments show that MMP14 plays a crucial role in the first stage of tendon development by detaching the collagen fibres from the fibripositors. MMP14 also promotes the formation of new collagen fibres; the tendons of mutant mice that lack MMP14 have fewer collagen fibres than normal mice. Further experiments revealed that the release of collagen fibres from fibripositors does not require MMP14 to cleave the chains of collagen proteins themselves. Instead, it appears that MMP14 cleaves another protein that is associated with the fibres, called fibronectin.

Taylor, Yeung, Kalson et al.'s findings show that MMP14 plays an important role in the development of tendons by releasing collagen fibres from fibripositors and promoting the formation of new fibres. The next challenge is to find out how MMP14 regulates the number of collagen fibres in mature tendons and other tissues, and how defects in this enzyme can lead to cancer and other diseases.

the fibril is located within actin-dependent (*Canty et al., 2006*) invaginations of the plasma membrane called fibripositors (*Canty et al., 2004*). Fibripositors are points of fibril assembly and sites of attachment of the cell to the ECM (*Kalson et al., 2013*) and exhibit a range of morphologies depending on the presence or absence of a slender finger-like projection of the plasma membrane. Protruding fibripositors exhibit the invagination and the finger-like projection whereas recessed fibripositors only exhibit the invagination (*Kalson et al., 2013*). The transport of collagen fibrils into fibripositors is powered by non-muscle myosin II and is not part of a fibril degradation process in embryonic tendon (*Kalson et al., 2013*). Short collagen fibrils can be found within membrane-sealed compartments termed fibricarriers (*Canty et al., 2004*; *Kalson et al., 2013*).

The transition from a unimodal distribution of narrow (~50 nm) diameter collagen fibrils in embryonic tendon to a bimodal distribution in adult tissues with means of ~50 nm and ~200 nm diameter fibrils is a fascinating phenomenon (*Parry et al., 1978*; *Derwin and Soslowsky, 1999*). The presence of narrow-diameter collagen fibrils in fibripositors during E14.5 to birth (in mice) signifies stage 1 of tendon development during which fibril number is determined (*Kalson et al., 2015*). Stage 2 occurs soon after birth (in mice) and is characterized by the disappearance of fibripositors, the release of fibrils to the ECM and the growth of fibrils in length and diameter (*Kalson et al., 2015*). We show here that in the absence of MMP14, progression from stage 1–2 does not occur, fibrils are retained by fibripositors, fibril diameters keep to ~50 nm, and tendon development stops at stage 1.

## Results

### *Mmp14*-deficient mouse tendons are thinner and mechanically weak

Wild type (WT) and *Mmp14* knockout (KO) littermates had similar birth weights but the KO mice were growth retarded at 3 days after birth (P3) (*Figure 1A*, as reported previously [*Holmbeck et al., 1999*]). Morphometric analyses at P0 showed that the tendons of the KO mice were thinner than those of WT mice (*Figure 1B*) and had fewer bundles of fibrils (*Figure 1C,D*). In transverse section, *Mmp14* KO tendons had fewer fibrils that were organized into fewer and irregular bundles (*Figure 1C,D*). We detected no difference in the number of cells per unit volume of the tendon between WT and *Mmp14* KO tendon (*Figure 1—figure supplement 1A–C*). There was no difference in fibril volume fraction (FVF) (*Figure 1D*); therefore, there was no evidence of abnormal fibril–fibril interactions. *Mmp14* KO tendons were mechanically weaker than WT tendon (*Figure 1E*). However, after adjustment for size, the KO tendons had similar mechanical properties to those of tendons from WT animals, which was consistent with the normal FVF. Quantitative PCR analysis showed no differences in *Col1a1* gene expression at P0 (*Figure 1F*) and [$^{14}$C]-proline labeling showed that procollagen processing was unaffected by loss of MMP14 (at P7, *Figure 1G*). However, despite no differences in gene expression or procollagen processing with MMP14 deficiency we found decreased collagen synthesis in KO samples, compared to WT animals (*Figure 1—figure supplement 1D*). We next analyzed collagen fibril diameters. As shown in *Figure 1H*, the diameters in P0 KO mouse-tail tendon were significantly larger (by ~12%) than those in WT mice.

### Conspicuous fibripositors in embryonic *Mmp14*-deficient cells

We used transmission electron microscopy (TEM), serial block face-scanning electron microscopy (SBF-SEM), and serial section electron tomography to examine embryonic WT and *Mmp14* KO tendons. We were careful to use tail tendon from anatomical site- and age-matched embryonic WT and KO mice. TEM analysis showed E15.5 WT tendons contained collagen fibrils in the ECM and a small number of collagen fibrils in fibripositors (*Figure 2A* and *Figure 2—figure supplement 1A*). In contrast, *Mmp14* KO tendons contained conspicuous electron-lucent invaginations characteristic of recessed fibripositors (*Figure 2B* and *Figure 2—figure supplement 1B*). There were ~8 times the number of fibripositor cross-sections per nucleus in KO tenocytes (*Figure 3—figure supplement 3C*). Tracing of fibrils in 3D reconstructions from SBF-SEM and electron tomography (*Figure 2—figure supplement 1C,D*) showed the presence of deep recessed fibripositors in *Mmp14* KO cells (*Videos 1, 2*, respectively) with looped collagen fibrils. *Figure 2—figure supplement 2* shows a diagrammatic representation of a typical recessed fibripositor containing looped collagen fibrils.

### No apparent collagen fibril transport abnormalities in *Mmp2* KO and *Mmp13* KO cells

MMP14 can activate proMMP2 (*Sato et al., 1994*) and proMMP13 (*Knauper et al., 1996*). Also, MMP2 inhibition blocked uptake and subsequent intracellular digestion of collagen fibrils in periosteal tissue explants (*Creemers et al., 1998*). However, EM analysis of embryonic *Mmp2* KO and *Mmp13* KO tail tendons showed no obvious changes to fibripositor occurrence (*Figure 3—figure supplement 1*).

### Fewer collagen fibrils in *Mmp14*-deficient embryonic tendons

We used SBF-SEM to quantitate the numbers and mean lengths of collagen fibrils in embryonic (E15.5) WT and *Mmp14* KO tendon, using methods described previously (*Starborg et al., 2013*). WT tendon contained numerous fibril tips and short fibrils (*Figure 3A*). In contrast, fibril tips were less frequent in KO tendon (*Figure 3A*). We then calculated the mean length of fibrils based on the relative frequency of tips-to-shaft numbers. The average fibril length in E15.5 WT tendon was 16 ± 3 μm whereas that in an age-matched and anatomical-site-matched *Mmp14* KO tendon was 38 ± 6 μm (*Figure 3B*). At E15.5, the tendon cross-sectional area and the FVF in WT and KO samples were not significantly different (*Figure 3C,D*). Therefore, given the same transverse area occupied by collagen fibrils in WT and KO tissue, the difference in collagen mean fibril length equates to a ~2.5-fold reduction in fibril number in KO tendon. Analysis at E16.5 confirmed that fibrils were, on average, shorter in WT tendon than KO tendon (mean length 50 ± 9 μm) compared with 111 ± 26 μm (n = 668 WT fibrils, 683 KO fibrils, respectively, each tracked over 10 μm).

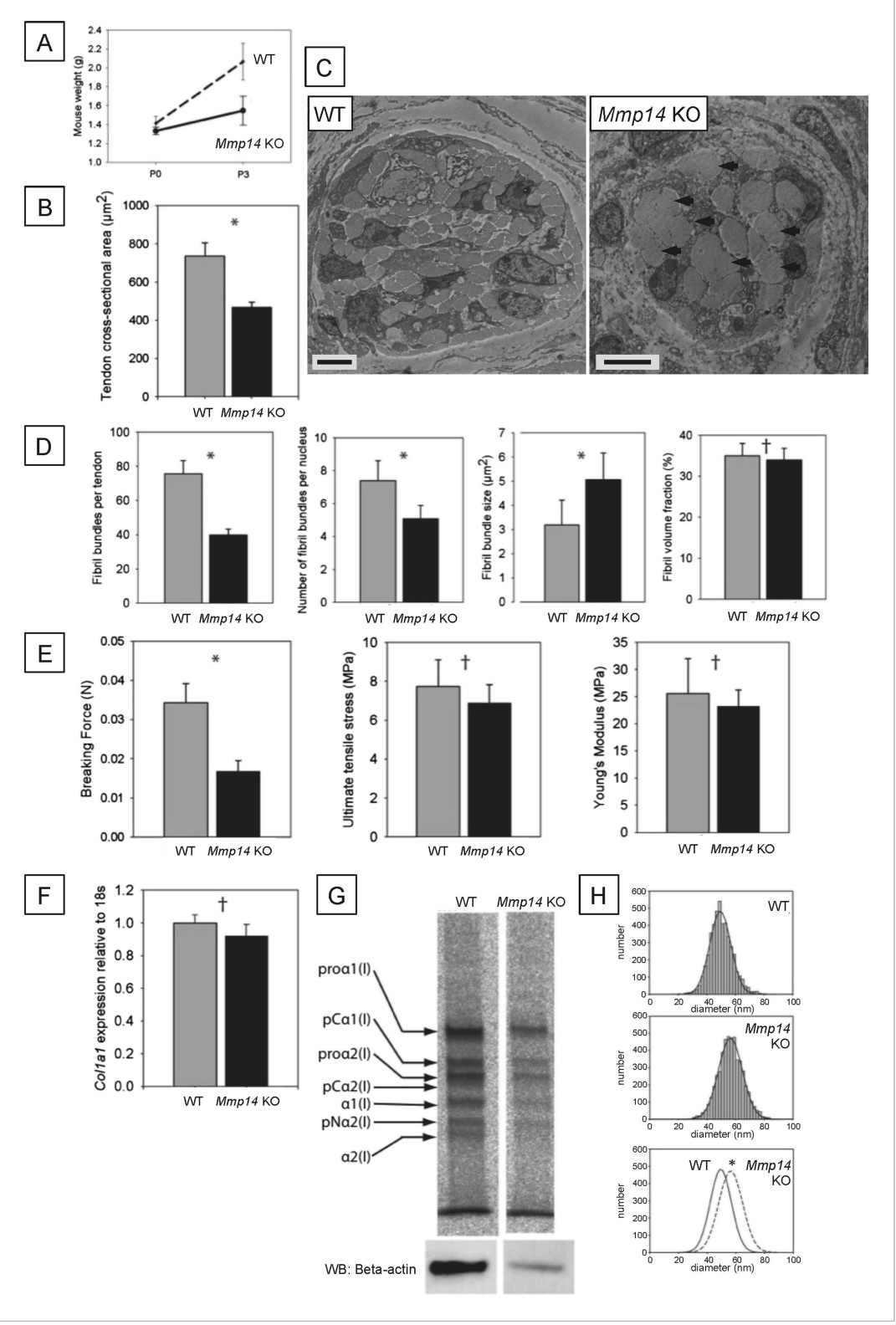

**Figure 1**. Neonatal *Mmp14*-deficient mice have small and weak tendons. (**A**) Weight of wild type (WT) and *Mmp14* knockout (*Mmp14* KO) littermates at birth (P0) and at 3 days postnatal (P3). (**B**) The cross-sectional transverse area of P0 *Mmp14* KO tendons is significantly smaller than WT tendons. (**C**) TEM images of P0 tail tendon demonstrate that KO tendons are smaller and show dysmorphic, enlarged bundles of collagen fibrils (arrowhead). Scale bars 5 μm.
*Figure 1. continued on next page*

*Figure 1. Continued*
(**D**) KO tendons have fewer, larger fibril bundles, but the FVF is not different to WT tendons. (**E**) KO tendons are weaker than WT tendons but have normal mechanical properties after adjusting for differences in size. (**F**) Analysis of *Col1a1* mRNA by qPCR in P0 tendons revealed no difference in gene expression. (**G**) $^{14}$C-proline labeling of collagen demonstrated normal collagen processing at P7 in WT and KO tail tendon. (**H**) Fibril diameter distributions of KO and WT tail tendon at P0 revealed significantly increased fibril diameters in KO mice. Bars show SEM. *p < 0.05, $^{†}$p > 0.05 (t-tests).
The following figure supplement is available for figure 1:

**Figure supplement 1**. Cell number and type I collagen synthesis in neonatal WT and *Mmp14* KO tail tendon.

## MMP inhibition causes abnormal collagen fibril transport

MMP14 has non-proteolytic activities (*Mori et al., 2013*). Therefore, we treated cells cultured in 3D tendon-like constructs (*Kapacee et al., 2008*) with the broad-spectrum MMP inhibitor GM6001 and performed TEM. Previous studies had confirmed that GM6001 was an effective inhibitor of MMPs in the tendon (*Kalson et al., 2013*). The addition of GM6001 to tendon-like constructs recapitulated the fibripositor phenotype of the *Mmp14*-deficient mouse (*Figure 3E,F*). This indicated that the catalytic activity of MMP14 is required for normal collagen fibril transport at fibripositors. Estimation of collagen mean fibril lengths in the constructs showed that inhibition of MMPs resulted in increased mean fibril length (*Figure 3G*), as was observed in embryonic *Mmp14* KO tendon.

## Newborn *Mmp14*-deficient mice have fewer collagen fibrils

SBF–SEM analysis showed that the mean length of fibrils in WT and *Mmp14* KO was the same in P0 tendons (*Figure 3—figure supplement 2*). This was in contrast to what was seen in embryonic tendon in which the fibrils were longer in KO tendons (*Figure 3*). Therefore, the more abundant but shorter fibrils in WT tendon grow in length during later embryonic development so that at P0 the fibrils are of equal mean length but there are more fibrils in the WT tendon than in *Mmp14* KO tendon.

## Embryonic *Col-r/r* mice have conspicuous fibripositors

The *Col-r/r* mouse carries a mutation in the MMP $^{3}/_{4}$-$^{1}/_{4}$ cleavage site in the triple helical domain of the α1 chain of type I collagen (*Liu et al., 1995*) that renders both the α1(I) and α2(I) chains resistant to cleavage by MMPs (*Wu et al., 1990*). Therefore, we were interested to compare collagen fibril transport in *Col-r/r* and *Mmp14* KO mice. In contrast to *Mmp14* KO, there were no significant differences in tendon size (*Figure 3—figure supplement 3A*) and collagen fibril diameter (*Figure 3—figure supplement 3B*) between embryonic (E15.5) WT and *Col-r/r* mice. Analysis of the frequency of fibricarrier profiles per nucleus (using methods used previously [*Canty et al., 2006*]) showed that *Mmp14* KO and *Col-r/r* tenocytes have significantly more fibripositor profiles than WT, with *Mmp14* KO cells having the most (*Figure 3—figure supplement 3C*). SBF–SEM analysis showed that *Col-r/r* tenocytes contained conspicuous recessed fibripositors, as seen in the *Mmp14* KO samples (*Figure 2C* and *Video 3*). Electron dense vacuoles were also present, which appeared to contain fibrillar structures in various stages of decomposition (*Figure 2C*, white arrows). Such compartments were rarely seen in WT or *Mmp14* KO samples.

## Stage 1 to 2 transition occurs in *Col-r/r* mice but not in *Scx-Cre::Mmp14 lox/lox* mice

We wanted to study the requirement of MMP14 on the stage 1 to stage 2 transition in the tendon but were unable to do so because the global *Mmp14* KO mice were distressed shortly after birth. Therefore, we generated a tendon-specific *Mmp14* KO mouse by crossing *Scleraxis-Cre* (*Scx-Cre*) mice with *Mmp14 lox/lox* mice (*Figure 4—figure supplement 1*). We examined the tendons at P0 by TEM and confirmed that the *Scx-Cre::Mmp14 lox/lox* tendons also contained multiple fibripositors with multiple fibrils, a similar phenotype observed for global *Mmp14* KO mice (*Figure 4A,B*). At birth the mice appear normal, however, the *Scx-Cre::Mmp14 lox/lox* mice exhibited a clear limb phenotype ~1 week after birth, with dorsiflexion of the fore and hind paws (*Figure 4C,D*, arrowhead) and a

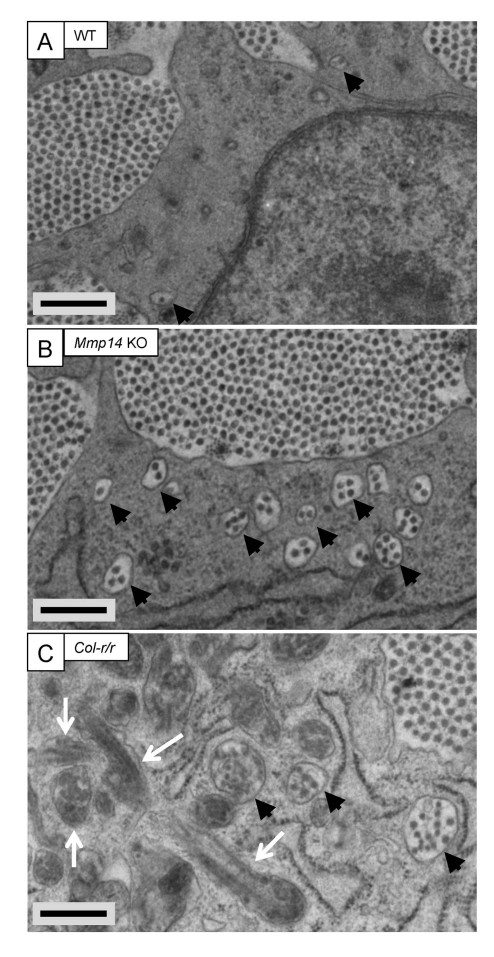

**Figure 2**. Fibricarrier analysis of wild-type, *Mmp14* KO, and *Col-r/r* embryonic tail tendon. Tail tendons at E15.5 of development from (**A**) wild-type, (**B**) *Mmp14* KO, and (**C**) *Col-r/r* mice. Black arrowhead, recessed fibripositor (electron lucent)-containing collagen fibrils. White arrow, enclosed electron-dense compartment. Scale bars 500 nm.

The following figure supplements are available for figure 2:

**Figure supplement 1**. *Mmp14*-deficient mice have prominent recessed fibripositors.

**Figure supplement 2**. Schematic showing looping of collagen fibrils in recessed fibripositors.

dome-shaped skull (*Figure 4D*, arrow). There was no apparent impairment in movement of tail or back muscles as observed in *Scx*-null mice (*Murchison et al., 2007*). Difference in size was apparent from 3 weeks postnatal concurrent with hip dysplasia and reduced bone density (*Figure 4—figure supplement 2A*), overgrowth of soft tissues in the paws (*Figure 4E*) and severe dorsiflexion of paws which was particularly obvious in the hind paws (*Figure 4F*). Smaller tendon (tail and Achilles) size and weaker bones were also observed in 7-week-old *Scx-Cre::Mmp14 lox/lox* mice. The mice became distressed from 7 weeks therefore analyses were performed no later than 7 weeks postnatal. Morphometric analysis of x-rays confirmed that *Scx-Cre::Mmp14 lox/lox* mice had significantly shorter cranium length and stunted skeletal growth (*Figure 4—figure supplement 2B*).

EM analysis of adult (7 weeks old) WT tendons showed the typical bimodal distribution of collagen fibril diameters (*Figure 5A*). Cells lacked fibripositors and were stellate in cross section (*Figure 5—figure supplement 1A*). In contrast, the tendons of *Scx-Cre::Mmp14 lox/lox* mice contained a unimodal distribution of small diameter fibrils (*Figure 5B*), and the cells were engorged with fibrils in membrane-bound compartments (*Figure 5—figure supplement 1B*). The inclusion of many fibrils into compartments made it difficult to delineate the cell–matrix interface. In comparison, cells in *Col-r/r* tendons were similar in appearance to those in WT tendons (*Figure 5—figure supplement 1C*) and the ECM contained fibrils with a broad bimodal distribution of diameters (*Figure 5C*).

## ImmunoEM shows MMP14 in recessed fibripositors

ImmunoEM of P7 (7 days postnatal) WT tendon using an anti-MMP14 antibody showed labeling of intracellular compartments without labeling the ECM or plasma membranes in contact with extracellular collagen fibrils (*Figure 5—figure supplement 2A*). As expected, *Mmp14*-null tendons were negative for MMP14 labeling (*Figure 5—figure supplement 2B*). Labeling using an anti-type I collagen antibody confirmed that the fibrils within the recessed fibripositors of *Scx-Cre::Mmp14 lox/lox* tendons contained type I collagen (*Figure 5—figure supplement 2C*).

## Fibronectin levels were consistently elevated in *Mmp14*-deficient tendons

To investigate why collagen fibrils were retained in fibripositors in *Mmp14*-null tendons, we used mass spectrometry LS/MS–MS to perform an unbiased comparison of proteins in *Mmp14*-deficient and WT

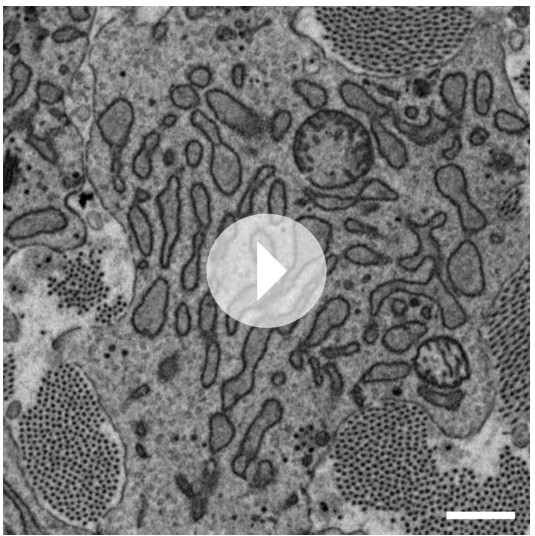

**Video 1.** Step-through video generated from SBF-SEM images of E17.5 embryonic WT mouse-tail tendon. z-depth is 100 µm. Scale bar 2 µm.

Achilles tendons. Care was taken to minimize muscle contamination and to remove as much associated loose connective tissue as possible. Although not quantitative, the analysis identified specific macromolecules that appear to be more abundant in *Scx-Cre::Mmp14 lox/lox* and global *Mmp14* KO tendons than in WT (*Supplementary file 1*). Peptides from FN were consistently more abundant in *Mmp14*-deficient samples. We identified a unique peptide from FN that was found in WT tendon in vivo and which contained an additional alanine residue at its N-terminus in *Mmp14*-deficient tendons (*Figure 6A*), suggesting that MMP14 is responsible for cleaving FN between Ala(1078) and Thr(1079). This was confirmed by LS/MS–MS analysis of recombinant human FN treated with recombinant human MMP14 prior to digestion with trypsin (data not shown). Immunofluorescence analysis at E15.5 showed accumulation of FN in *Scx-Cre::Mmp14 lox/lox* tendons compared to WT tendons and the levels of FN appeared to progressively accumulate in KO tendons at P0 and P10 (*Figure 6B*). The LS-MS/MS analyses also identified periostin and integrins including α11β1 (*Supplementary file 1*). Analysis of periostin in tendons showed similar intensities at E15.5 and P0 but was increased in the tendon epithelium at P10 *Scx-Cre::Mmp14 lox/lox* mice compared to WT mice (*Figure 6—figure supplement 1A*). Subsequent western blot analysis of P7 mice confirmed that levels of FN were higher in *Scx-Cre::Mmp14 lox/lox* tendons compared to WT littermates (*Figure 6C,D*). We stripped and re-probed the blot for periostin detection and confirmed that it was not accumulated to the extent FN was in *Scx-Cre::Mmp14 lox/lox* tendons (*Figure 6—figure supplement 1B*).

Next, we wanted to determine if elevated levels of FN might account for the fibripositor phenotype and retention of collagen fibrils at the cell surface. Thus, we formed tendon-like constructs in the presence of 200 µg/ml exogenous human plasma FN. The constructs formed within ~10 days as previously described (*Kapacee et al., 2008*). TEM of the constructs showed pronounced fibripositors in cells incubated in exogenous FN (*Figure 6—figure supplement 2*).

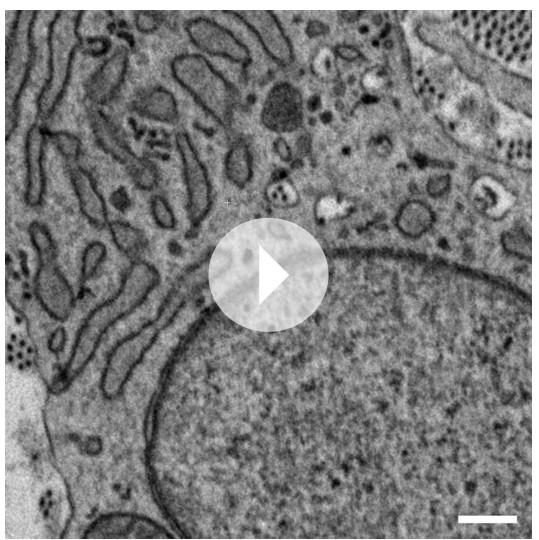

**Video 2.** Step-through video generated from SBF-SEM images of E17.5 embryonic *Mmp14* KO mouse-tail tendon. z-depth is 100 µm. Scale bar 2 µm.

## Discussion

We show here that MMP14 is essential for the stage 1–stage 2 transition of tendon development (which occurs around birth in the mouse) by catalyzing the release of collagen fibrils from fibripositors. In WT mouse tendons, fibripositors disappear and collagen fibrils are released to the ECM soon after birth (marking the end of stage 1), and the fibrils grow in diameter and length (marking the start of stage 2) (*Kalson et al., 2015*). In the absence of MMP14, the fibrils are retained within fibripositors and the number of collagen fibrils formed during stage 1 is reduced. As a result, tendons in *Mmp14*-deficient mice are thinner compared to WT. Although MMP14 is capable of cleaving type I collagen at the $^{3}/_{4}$-$^{1}/_{4}$

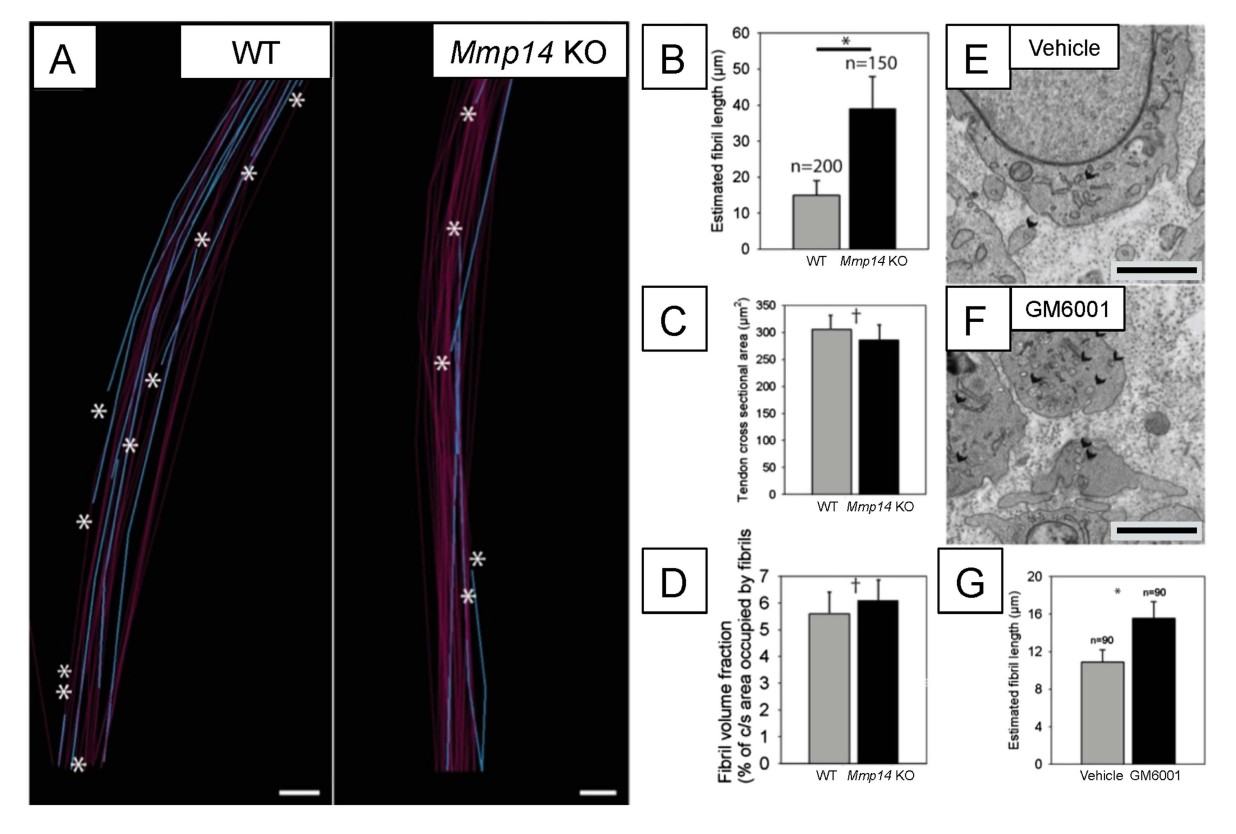

**Figure 3.** Deficiency in MMP14 activity results in fewer collagen fibrils. (**A**) 10 μm-deep (z-axis) slices of 3D reconstructions of SBF-SEM data taken from of WT and *Mmp14* KO embryonic tendons at E15.5 showing collagen fibrils (blue) with a tip (marked by asterisks) found within the volume. Purple fibrils passed through the volume and so did not have tips in the reconstruction. Scale bars 500 nm. (**B**) Quantification of mean fibril length based on the number of tips identified shows that E15.5 WT fibrils are shorter than fibrils in age- and anatomical position-matched tail tendons from KO mice (308 and 266 fibrils tracked, respectively). (**C**) Tendon cross-sectional area and (**D**) FVF are not different at E15.5 KO tendons. (**E**, **F**) Electron microscopy of tendon-like constructs cultured in the presence of MMP inhibitor GM6001 (10 μM in 0.1% DMSO) show increased number of recessed fibripositors (arrowheads) compared to vehicle control. Scale bars 1 μm. (**G**) Increase in calculated mean fibril length in GM6001-treated tendon-like constructs. Bars show SEM. *p < 0.05, †p > 0.05 (t-tests).

The following figure supplements are available for figure 3:

**Figure supplement 1**. Embryonic tendons deficient in *Mmp2* or *Mmp13* do not have overt tendon phenotypes.

**Figure supplement 2**. Deficiency in MMP14 activity results in fewer collagen fibrils tips at P0.

**Figure supplement 3**. Quantitative analysis of *Col-r/r* embryonic tendons.

helical site, we show that cleavage at this site is not required for tendon development. Interestingly, FN accumulates in *Mmp14*-deficient tendons. Thus, we propose that the ability of MMP14 to cleave macromolecules other than type I collagen is essential for releasing collagen fibrils from fibripositors.

Collagen fibrils are assembled on the surface of embryonic tenocytes and pulled into fibripositors by a mechanism powered by non-muscle myosin II (*Kalson et al., 2013*). The absence of fibripositors in stage 2 shows that fibrils are 'released' from the plasma membrane for delivery to the ECM (*Kalson et al., 2015*). We propose that collagen fibrillogenesis in embryonic tendon occurs via an 'APR' mechanism of collagen 'attachment' to the cell surface, non-muscle myosin II-powered 'pulling' on fibrils into fibripositors, and MMP14-mediated 'release' of the fibrils to the ECM.

*Mmp14*-null mice had thinner tendons than WT mice. SBF-SEM analyses showed similar cell numbers in embryonic WT and null mouse tendons; therefore, we excluded the possibility that

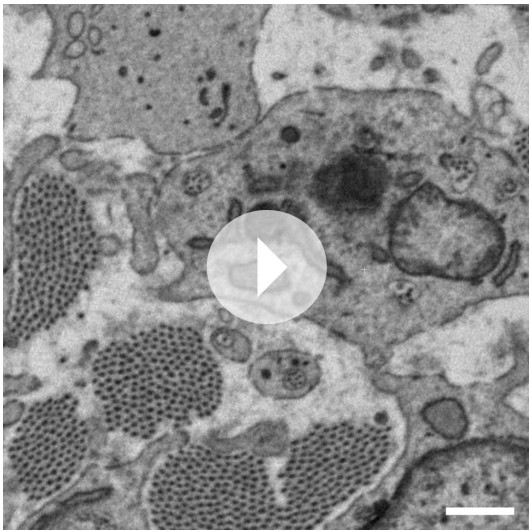

**Video 3.** Step-through video generated from SBF-SEM images of E17.5 embryonic *Col-r/r* mouse-tail tendon. z-depth is 100 μm. Scale bar 2 μm.

delayed development was the cause. However, *Mmp14*-null tendons had fewer collagen fibrils. As shown previously, the lateral size of the tendon is established during embryonic development when embryonic tenocytes assemble a finite number of collagen fibrils at fibripositors (*Kalson et al., 2015*). The fibrils are released after birth and subsequently grow in length and diameter in a process of matrix expansion. Therefore, the fewer fibrils in *Mmp14*-deficient tendons appear to be a direct result of the inability of the cells to release collagen fibrils to the ECM before a new cycle of fibril assembly can begin. In the presence of continued collagen synthesis (albeit at a reduced rate [*Figure 1—figure supplement 1D*]), the existing fibrils continue to grow in length and diameter at the expense of nucleation of new fibrils. At P0 and soon after birth, the fibrils in WT tendon grow in length to equal the length of fibrils in *Mmp14* KO tendons.

Tendon size was normal in *Col-r/r* embryos; therefore, the absence of type I collagen cleavage was not the cause of reduced fibril number in *Mmp14*-deficient tendons. However, tenocytes in embryonic *Col-r/r* tendons contained electron-dense vacuoles, which were morphologically similar to those previously observed (*Beertsen et al., 1978*; *Everts et al., 1996*). Therefore, although type I collagen is cleaved at the $^{3}/_{4}$-$^{1}/_{4}$ site during embryonic development, cleavage is not essential for tendon development. While the *Col-r/r* mutation renders the triple helix resistant to cleavage by MMPs, rodent MMP13 recognizes an additional cleavage site C-terminal to the N-telopeptide crosslink (*Krane et al., 1996*). *Liu et al. (1995)* reported that the MMP13 cleavage site in type I collagen permits normal remodeling during development and early postnatal life, but that cleavage at the $^{3}/_{4}$-$^{1}/_{4}$ site is needed for subsequent remodeling and accounts for the observed progressive marked skin fibrosis in the *Col-r/r*. Our data agree with these conclusions and show that MMP13 (and MMP2) is not essential for tendon development. The fact that degradative vacuoles are rare in embryonic *Mmp14* KO tenocytes and that fibril numbers were reduced and fibril lengths are greater in *Mmp14* KO cells, suggests that the vacuoles might be part of a mechanism to regulate fibril number and/or length.

Two proteins, FN and periostin, stood out in the LC-MS/MS comparison of WT and *Mmp14*-null tendons as proteins that could help to explain the *Mmp14*-null tendon phenotype (*Supplementary file 1* shows number of peptides from periostin and FN were over-represented in *Mmp14* KO tendon). Periostin is a member of the matricellular family of secreted proteins that modulate cell–ECM interactions (*Sage and Bornstein, 1991*; *Murphy-Ullrich and Sage, 2014*). Periostin is highly expressed by epithelial cells, is bound by αvβ3 and αvβ5 integrins, is upregulated in epithelial tumors to support adhesion and migration (*Gillan et al., 2002*; *Yuyama et al., 2002*; *Liu and Du, 2015*), and is a prognostic marker for TH2-driven asthma (*Parulekar et al., 2014*) and lung fibrosis (*Amara et al., 2015*). Additional studies have shown that periostin supports tendon formation in an ectopic mouse model of the development of tenogenic tissue (*Noack et al., 2014*). Evidence also suggests that periostin interacts with type I collagen to regulate collagen fibrillogenesis (*Noack et al., 2014*). It has also been reported that periostin deficiency might cause collagen fibril disorganization and affect the distribution of FN (*Tabata et al., 2014*). We showed that periostin was increased in the tendon epithelium (*Taylor et al., 2011*) that surrounds the body of the tendon, in P10 *Scx-Cre::Mmp14 lox/lox* mice compared to WT mice (*Figure 6—figure supplement 1A*). Periostin can be cleaved by MMP14 in vitro (*Stegemann et al., 2013*) and therefore its accumulation in the epithelium could be a direct result of substrate accumulation. It is also possible that the elevated levels seen in the epithelium are an indirect result of MMP14 deficiency in the fibrous core of the tendon.

We observed elevated levels of FN in *Mmp14* KO tendon, as shown by LC-MS/MS and by immunofluorescence. MMP14 has been shown to cleave several macromolecules in vitro including FN

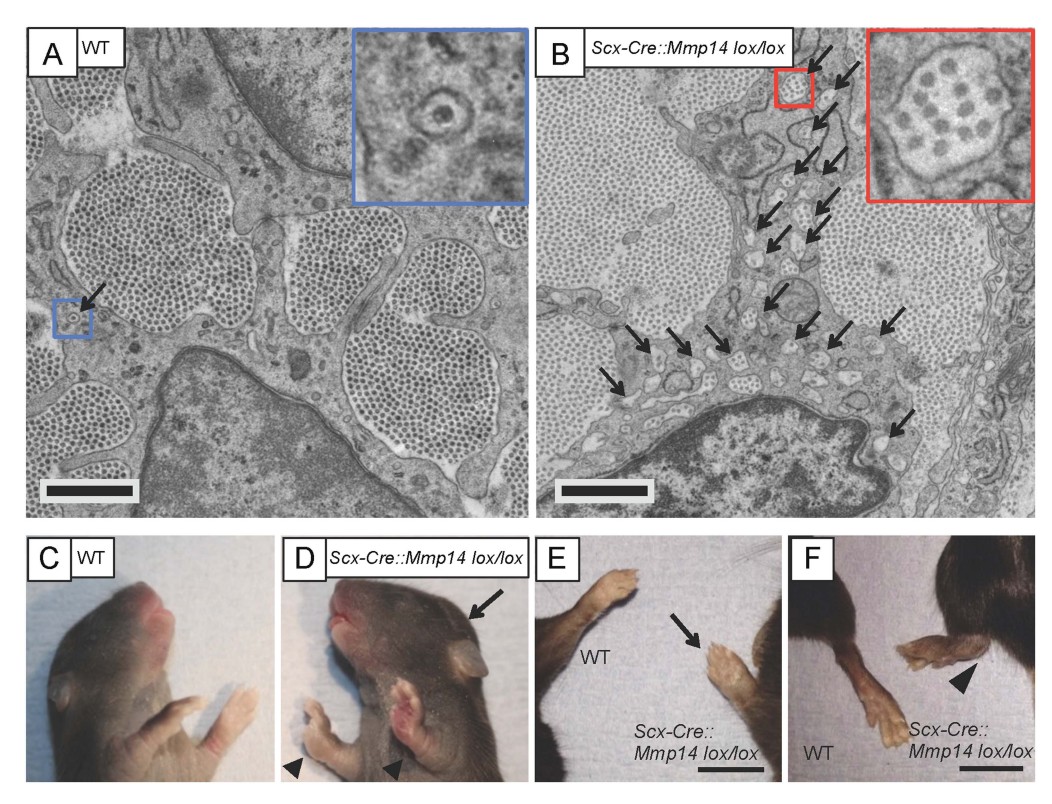

**Figure 4**. *Scx-Cre::Mmp14 lox/lox* mice have limb and skeletal deformities. Tail tendons from littermates at P0 from (**A**) WT and (**B**) *Scx-Cre::Mmp14 lox/lox* mice show *Mmp14*-null tendons have multiple fibripositors-containing multiple fibrils (red box) than fibripositors in WT tendons (blue box). Black arrowhead, recessed fibripositor (electron lucent)-containing collagen fibrils. Scale bars 500 nm. (**C**) Control pups at P8 showed normal limb development but (**D**) *Scx-Cre::Mmp14 lox/lox* littermates show dorsiflexion of their limbs (arrowhead) and dome-shaped skull (arrow). Adult (7 week old) *Scx-Cre::Mmp14 lox/lox* mice have (**E**) enlarged paws (open arrow) and (**F**) extreme dorsiflexion of hind limbs (arrowhead) compared to control littermates. Scale bars 1 cm.

The following figure supplements are available for figure 4:

**Figure supplement 1**. Genotyping the *Scx-Cre::Mmp14 lox/lox* colony.

**Figure supplement 2**. Adult *Scx-Cre::Mmp14 lox/lox* mice have skeletal deformities.

(*d'Ortho et al., 1997*; *Ohuchi et al., 1997*; *Tam et al., 2004*; *Butler and Overall, 2007*); therefore, the accumulation of FN in *Mmp14*-deficient tendon might be a direct result of the absence of MMP14. We observed a potential cleavage site between Ala(1078) and Thr(1079) in FN that occurred in WT but not in *Scx-Cre::Mmp14 lox/lox* tendon. FN is a core component of extracellular matrices (*Mao and Schwarzbauer, 2005*) and has an important role in development (*George et al., 1993*) and wound healing (*Sakai et al., 2001*). Mice lacking FN are embryonic lethal with defects in mesoderm formation (*George et al., 1993*). Furthermore, FN co-distributes with type I and III collagen (*Fleischmajer and Timpl, 1984*). To understand if the presence of elevated levels of FN could explain the fibripositor phenotype, we incubated tendon-like constructs with exogenous FN. This resulted in a profound increase in appearance of fibripositors. Taking the EM, LC-MS/MS, and tendon-construct data together, we propose that FN forms a 'molecular bridge' between the cell and the collagen fibril. Thus, cleavage of the bridge and removal of fibripositors triggers the onset of stage 2 of tendon development and subsequent expansion of the matrix.

An unexpected observation was the effect on skeletal size of deleting MMP14 from tendon. The shortened long bones in *Scx-Cre::Mmp14 lox/lox* mice suggests important consequences of tendon

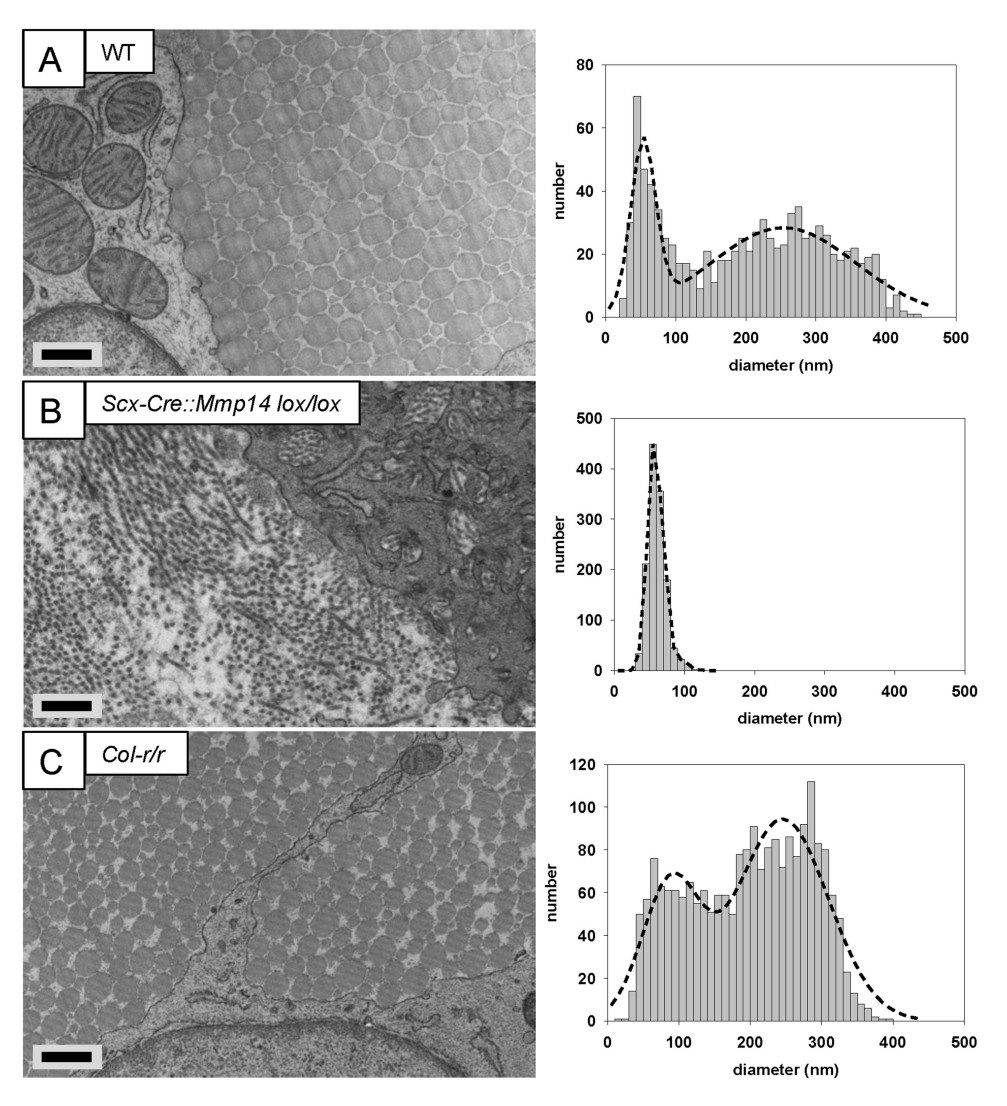

**Figure 5**. Deficiency in MMP14 activity inhibits bimodal fibril diameter distribution in tendons from adult mice. Tail tendons from 7 week-old (**A**) WT, (**B**) *Scx-Cre::Mmp14 lox/lox*, and (**C**) *Col-r/r* mice. Larger diameter fibrils can be observed in the ECM of WT and *Col-r/r* postnatal tendons but only narrow diameter fibrils are observed in *Mmp14*-deficient postnatal tendons. Scale bars 500 nm.

The following figure supplements are available for figure 5:

**Figure supplement 1**. Cleavage of the $^3/_4$-$^1/_4$ collagen-I site is not required for release of fibrils in tendons from adult mice.

**Figure supplement 2**. Immuno-electron microscopy of *Scx-Cre::Mmp14 lox/lox* tendon.

development on skeletal growth. LC-MS/MS analyses showed under-representation of decorin, COMP, PCOLCE, and TNXB in *Mmp14*-null tendons. These proteins are directly involved in regulating collagen fibril size and shape; mice lacking decorin have fibrils with irregular outlines (*Danielson et al., 1997*), COMP can act as a catalyst for collagen fibril formation (*Halasz et al., 2007*), PCOLCE enhances the cleavage of procollagen to collagen (*Takahara et al., 1994*) and mice lacking TNXB have reduced numbers of collagen fibrils (*Mao et al., 2002*). In contrast, several proteins were over-represented in *Mmp14*-deficient tendons. These included filamin A, which has functions in cell–ECM adhesion (*Nakamura et al., 2011*) and mechanotransduction (*Jahed et al., 2014*). As tendons are

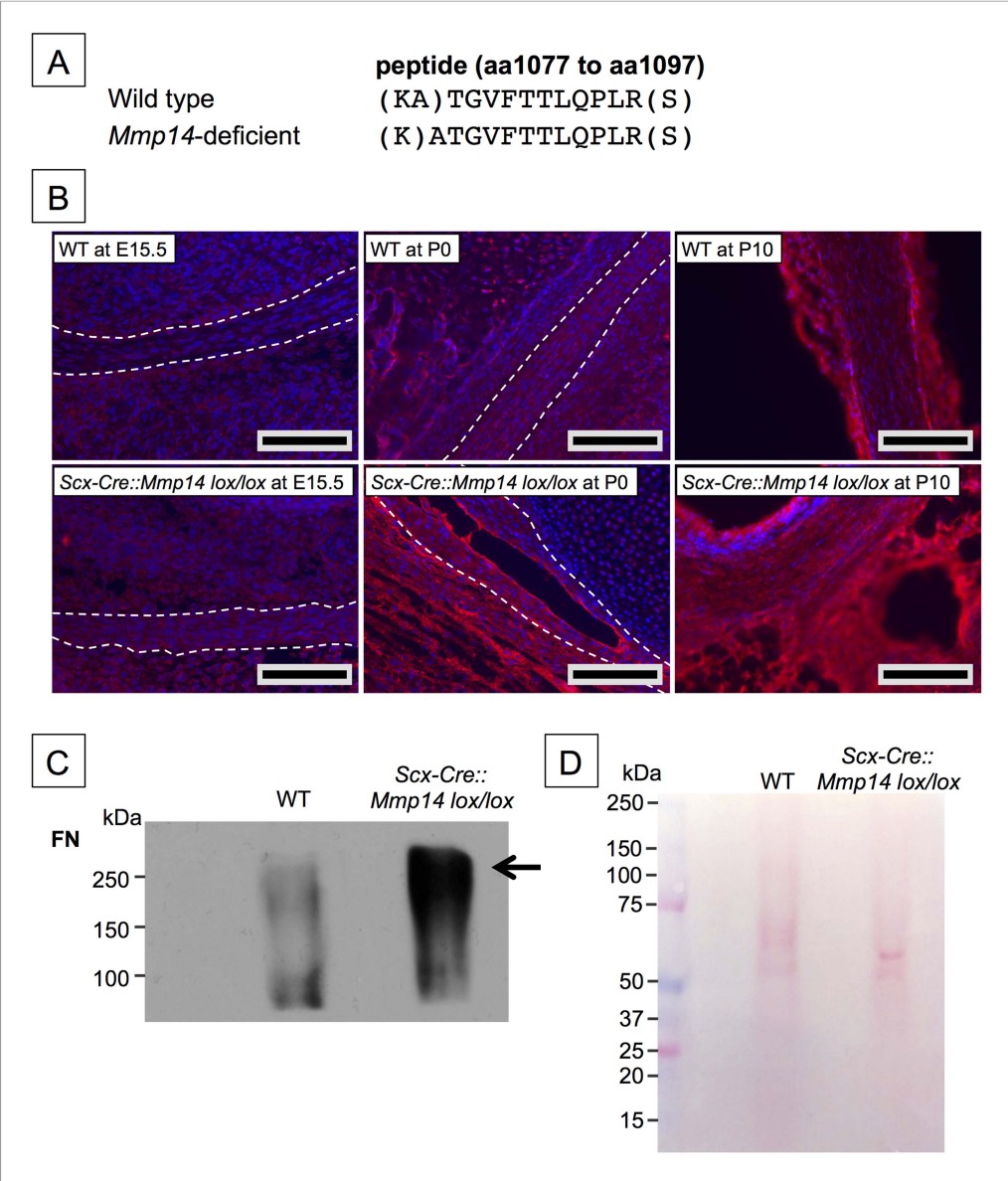

**Figure 6.** Elevated FN in *Mmp14*-deficient tendons. (**A**) Sequence of a unique semi-tryptic peptide of FN identified in neonatal (P7-10) WT tendon and the sequence of the corresponding peptide from *Mmp14*-deficient tendons without the additional Ala(1078)-Thr(1079) cleavage. (**B**) Immunofluorescence analysis of FN in tendons of WT and *Scx-Cre::Mmp14 lox/lox* mice at E15.5, P0, and P10 of development. Scale bars 200 μm. (**C**) Western blot analysis of P7 WT and *Scx-Cre::Mmp14 lox/lox* tendons show elevated FN in *Mmp14*-deficient tendons. (**D**) Ponceau S-stained membrane shows equivalent extractability of WT and *Scx-Cre::Mmp14 lox/lox* tendons.

The following figure supplements are available for figure 6:

**Figure supplement 1**. Elevated periostin levels only in postnatal *Scx-Cre::Mmp14 lox/lox* tendons.

**Figure supplement 2**. Exogenous FN induces recessed fibripositors in tendon-like constructs.

predominately ECM, changes in ECM composition and cell–ECM interactions are likely to have profound effects on cell signaling (e.g., ECM growth factor presentation) as well as the growth and mechanical properties of tendon leading to changes in musculoskeletal development. Finally, the tendons, cartilage, muscle, and bone are peripheral circadian clocks, each with their unique circadian

transcriptome (see [*Yeung et al., 2014a*] and reviewed by *Dudek and Meng (2014)*). Therefore, changes in the organization and mechanical properties of the tendon might affect its circadian entrainment and that of adjacent musculoskeletal tissues.

## Materials and methods

### Animals

The care and use of all mice in this study was carried out in accordance with UK Home Office regulations, UK Animals (Scientific Procedures) Act of 1986 under the UK Home Office licence (PPL 40/3485). All animals were sacrificed by a Schedule 1 procedure by trained personnel. *Mmp14* KO mice were as described previously (*Zhou et al., 2000*). To generate mice in which *Mmp14* is ablated in tendon-lineage cells, we crossed mice-expressing Cre recombinase under the control of Scleraxis (Scx-Cre; C57BL/6) (*Blitz et al., 2013*) with mice carrying the floxed exons (exons 2 to 4) of the *Mmp14* gene (*Mmp14 lox/lox*; C57BL/6) (*Zigrino et al., 2012*). *Mmp13* KO embryos were a generous gift from Zena Werb (*Stickens et al., 2004*). *Mmp2* heterozygous mice were imported from RIKEN BioResource Center (GelAKO/RBRC00398; C57) (*Itoh et al., 1997*) and bred to homozygosity. *Col-r/r* mice were imported from Jackson Laboratory (B6;129S4-Col1a1tm1Jae/J) (*Liu et al., 1995*). X-ray analyses were performed as described previously (*Yeung et al., 2014b*).

### Mechanical testing

The methods used were as described previously (*Kalson et al., 2010*). Tendon (from tail) diameters were measured from digital photographs. The diameter, d, was then used to calculate transverse area according to the formula $\pi d^2/4$. This assumed a circular transverse shape as used in mechanical testing of tissue engineered ligament (*Hairfield-Stein et al., 2007*). An average of three diameter measurements was recorded for each tendon. The original contour length of tendons was measured from a digital photograph of the mounted construct. A tare load of 10 mN was applied at the start of the tensile test to fully straighten the tendon. The length at failure was determined from the Instron test (giving change in length L$\Delta$). The tendons were tested to failure with a strain rate of 5 mm per minute (equivalent to approximately 1% strain per second).

### Electron microscopy

A minimum of 3 tail tendons was examined for each experiment. The tendons were prepared for TEM and SBF-SEM as described (*Starborg et al., 2013*), with care being taken to maintain the length and tension during fixation. Sections (70-nm thick) were examined for TEM using an FEI Tecnai 12 instrument fitted with a 2k × 2k-cooled CCD camera (F214A, Tietz Video and Image Processing Systems, Gauting, Germany). Serial section electron tomography was completed as described using semi-thick (300 nm) serial sections were collected on formvar-coated copper slot grids. Orthogonal tilt series were then acquired on a FEI Tecnai Polara TEM operated at 300 kV (*Kalson et al., 2013*). Tomograms were generated and contours modeled in IMOD (*Kremer et al., 1996*). The methods used for SBF-SEM were as described (*Starborg et al., 2013*) using a Gatan 3View microtome within an FEI Quanta 250 scanning microscope. Cell number measurements were made on 3 separate SBF-SEM samples for WT and *Mmp14* KO tail tendons at P0. The volume of the tendon tissue in each SBF-SEM 3D reconstruction was calculated and all the cells in the volume were reconstructed using IMOD. Each cell nucleus contained within the reconstruction was identified and counted. Cells per 1000 $\mu m^3$ of tissue were calculated to allow comparison between samples. ImmunoEM was performed as previously described using high-pressure freezing and freeze substitution into LR White resin (*Canty et al., 2004*). A rabbit anti-Collagen-I antibody (T40777R; Meridian Life Science, Inc.) and a mouse anti-MMP14 antibody (MAB3328; Merk Millipore) were used.

### Mass spectrometry

Cleanly dissected neonatal (P7-10) mouse Achilles tendons were snap frozen in liquid nitrogen and disrupted in 0.1 M Tris, pH 7.5 using a B Braun Mikro-dismembrator S (2 × 90 s, 2000 rpm). Tissue samples were digested with trypsin (12.5 ng/$\mu$l; Sigma) overnight at 37°C in 25 mM ammonium bicarbonate (pH 7.5). Human rhFurin (2 ng/ml) in 100 $\mu$l activation buffer (50 mM Tris-HCl, 1 mM CaCl$_2$, 0.5% Brij-35, pH 9) was added to 4 $\mu$l human rhProMMP14 (0.37 $\mu$g/$\mu$l) and incubated for 1.5 hr

at 37°C. Recombinant human FN (1 µg/ml) was incubated with activated rhMMP14 (1 ng/ml) at 37°C for 1 hr. For gel-top analysis, MMP14-treated FN was briefly separated by electrophoresis under reducing conditions. A gel top band was excised from the stained gel and processed using in-gel tryptic digestion.

For in-gel tryptic digestion, proteins were excised from SDS-PAGE gels and dehydrated using acetonitrile followed by vacuum centrifugation. Dried gel pieces were reduced with 10 mM dithiothreitol and alkylated with 55 mM iodoacetamide. Gel pieces were then washed alternately with 25 mM ammonium bicarbonate followed by acetonitrile and dried by vacuum centrifugation. Samples were digested with trypsin, as above. Digested samples were analyzed by LC-MS/MS using an UltiMate 3000 Rapid Separation LC (RSLC, Dionex Corporation, Sunnyvale, CA) coupled to an Orbitrap Elite (Thermo Fisher Scientific, Waltham, MA) mass spectrometer. Peptide mixtures were separated using a gradient from 92% A (0.1% FA in water) and 8% B (0.1% FA in acetonitrile) to 33% B, in 44 min at 300 nl/min, using a 250 mm × 75 µm i.d. 1.7 µM BEH C18, analytical column (Waters). Peptides were selected for fragmentation automatically by data-dependent analysis.

## Mass spectrometry data analysis

Data produced were searched using Mascot (Matrix Science UK), against *uniprot* with taxonomy of *Mus musculus* selected. Cysteine carbamidomethylation was selected as a fixed modification, and lysine and proline oxidation included as variable modifications, for all enzymes. For trypsin, methionine oxidation was additionally included as a variable modification and the enzyme selected as 'semi-trypsin'. Data were validated using Scaffold (Proteome Software, Portland, OR).

## Western blot analysis

Mouse Achilles tendons (at P7) were dissected clean of contaminating the muscle, snap frozen in liquid nitrogen and disrupted using a B Braun Mikro-dismembrator S (2 × 90 s, 2000 rpm). Proteins were extracted directly into RIPA buffer (50 mM Tris, pH 7.6, 150 mM NaCl, 0.1% SDS, 1 mM EDTA and 1% NP-40) containing EDTA-free protease inhibitor cocktail (Roche) and quantified using a BCA assay. Samples (100 µg) were reduced and analyzed by western blotting and densitometry. Western blots were stripped with 2% SDS, 62.5 mM Tris HCl pH 6.8 and 100 mM 2-mercaptoethanol for 50 min at 50°C. Fibronectin (FN) was detected using a rabbit polyclonal antibody (ab2413; Abcam), and periostin was detected using a goat polyclonal antibody (AF2955; R&D Systems).

## Immunofluorescence

From WT and *Scx-Cre::Mmp14 lox/lox* mice, whole hind limbs from E15.5 embryos, lower hind limbs without the skin from P0 pups and dissected Achilles tendons from P10 pups were cryo-preserved in OCT-embedding matrix (Thermo Scientific). Longitudinal sections of 8-µm thickness were fixed with 4% paraformaldehyde in PBS for 15 min, permeabilized with 0.2% Triton X-100 in PBS for 10 min and then blocked with 2% BSA in PBS for 1 hr. FN was detected using a rabbit polyclonal antibody (ab23750; Abcam; diluted 1:500), and periostin was detected using a goat polyclonal antibody (ab14041; Abcam; diluted 1:500). Cy3-conjugated secondary antibodies (Invitrogen) were used and sections were mounted using Vector Shield containing DAPI (Vector Laboratories). Fluorescent images were taken using a digital camera attached to an Olympus BX51 and captured using MetaVue imaging software (Molecular Devices).

## $^{14}$C labeling of type I collagen in the tendon

P7 tail tendons were labeled with $^{14}$C-proline for 1 hr and separate extracellular and intracellular extracts prepared as described (*Canty et al., 2004*). The intracellular extract was analyzed by electrophoresis using 4–12% pre-cast Bis-Tris gels and MES running buffer. The gels were divided and the top half of the gel (>70 kDa) fixed, dried, and analyzed by autoradiography to detect $^{14}$C-collagen. The bottom half of the gel (<70 kDa) was analyzed by Western blotting with an antibody to β-actin; signal was detected using a CCD camera system. The gels were analyzed by

densitometry with the relative band intensities determined by comparison to serial dilutions of an independent sample.

## Acknowledgements

We thank Ronen Schweitzer for the *Scx-Cre* mice. We thank Vincent Everts, Yoshi Itoh and Karl Tryggvason for scientific discussions, as well as Vicky Taylor, Alison Hallworth and Raymond Hodgkiss and the University of Manchester Biological Support Facility for animal welfare and husbandry, and the staff of the Bio-MS Core Research Facility in the Faculty of Life Sciences, University of Manchester. The work was generously funded by the Wellcome Trust (091840/Z/10/Z to KK). CM and PZ were supported by Deutsche Forschungsgemeinschaft through SFB 829 (B4, CRC829).

## Additional information

### Funding

| Funder | Grant reference | Author |
| --- | --- | --- |
| Wellcome Trust | 091840/Z/10/Z | Karl E Kadler |
| Deutsche Forschungsgemeinschaft (DFG) | B4, CRC829 | Cornelia Mauch |

The funders had no role in study design, data collection and interpretation, or the decision to submit the work for publication.

### Author contributions

SHT, YL, TS, SW, DFH, Acquisition of data, Analysis and interpretation of data; C-YCY, Acquisition of data, Analysis and interpretation of data, Drafting or revising the article; NSK, EGC-L, Conception and design, Acquisition of data, Analysis and interpretation of data; PZ, CM, Analysis and interpretation of data, Contributed unpublished essential data or reagents; KEK, Conception and design, Analysis and interpretation of data, Drafting or revising the article

### Author ORCIDs

Nicholas S Kalson, http://orcid.org/0000-0001-8394-3060
Karl E Kadler, http://orcid.org/0000-0003-4977-4683

### Ethics

Animal experimentation: The care and use of all mice in this study was carried out in accordance with UK Home Office regulations, UK Animals (Scientific Procedures) Act of 1986 under the UK Home Office licence (PPL 40/3485). All animals were sacrificed by a Schedule 1 procedure by trained personnel. MMP14 KO mice were as described previously (Zhou et al., 2000). To generate mice in which MMP14 is ablated in tendon-lineage cells, we crossed mice expressing Cre recombinase under the control of Scleraxis (ScxCre; C57BL/6) (Blitz et al., 2013) with mice carrying the floxed exons (exons 2 to 4) of the MMP14 gene (MMP14-floxed; C57BL/6) (Zigrino et al., 2012). MMP13 KO embryos were a generous gift from Zena Werb (Stickens et al., 2004). MMP2 heterozygous mice were imported from RIKEN BioResource Center (GelAKO/RBRC00398; C57) (Itoh et al., 1997) and bred to homozygosity. Col-r/r mice were imported from Jackson Laboratory (B6;129S4-Col1a1tm1Jae/J) (Liu et al., 1995). X-ray analyses were performed as described previously (Yeung et al., 2014).

## Additional files

### Supplementary file

• Supplementary file 1. LC-MS/MS peptide identity and counts. Editable Microscoft Excel sheet. Peptide identity was ≥99%. Color codes show runs performed from tendons isolated at the same time. Data for fibronectin and periostin are highlighted in yellow.

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
