## [Decision Letter]

Thank you for submitting your work entitled “Matrix metalloproteinase 14 is required for fibrous tissue expansion” for peer review at *eLife*. Your submission has been favorably evaluated by Fiona Watt (Senior Editor), a Reviewing Editor (Robb Krumlauf), and three reviewers.

The reviewers have discussed the reviews with one another and Robb Krumlauf, as the Reviewing editor, has drafted this decision to help you prepare a revised submission.

The consensus opinion of the reviewers is that this is an important new study analyzing the role of MMP14 during tendon development, which represents a significant contribution to our understanding of collagen fibril synthesis, release and growth. The manuscript provides a detailed analysis of the MMP14 (MT1-MMP) knockout phenotype in the tendon. They show that *Mmp14*-null mice have reduced tendon cross sectional area and fewer fibrils at birth. They also observed an increase in fibril diameter and fibripositor number. This is specific to MMP14 as other MMP knockouts do not phenocopy these characteristics. The phenotype does not appear to be due to MMP cleavage of collagen as examination of a cleavage insensitive collagen knockout does not phenocopy the MMP14 results. Fibronectin amounts were increased in MMP14 deficient tendons and in vitro assays demonstrated that incubation with fibronectin caused more pronounced fibripositors. This leads to the novel conclusion that MMP14 is necessary for tendon growth by mediating the release of collagen into the ECM.

An interesting model is presented with a plausible explanation for their findings. In the model MMP14 functions in collagen release from the fibripositors. As this presents a fascinating mechanism for how a fibrous matrix is assembled, it would be of general interest to the field of tendon biology. However, there are several key discrepancies and issues of clarity relating to the data that need to be addressed. Hence, the work requires major revision as multiple findings presented in the paper appear disjointed or lack sufficient data to fully support the main conclusions. These major issues are: 1) The authors outline two stages of tendon development: in the first stage (from E14.5-) collagen fibril number is increased, and the second stage, at early postnatal stages, the fibrils are released to the ECM where they grow in length and diameter, and fibripositors disappear. They conclude that MMP14 is required for the progression from stage 1 to stage 2, affecting tissue expansion. However, their data suggest that MMP14 may act at multiple stages of tendon development. There is a lack of clarity on the phenotype and stages analyzed for the *Scx-Cre-Mmp14* tissue specific knockout compared with the null. Much of their data (Figures 1, 2 and 3) are on the phenotype of the *Mmp14*-null mice at embryonic and early postnatal stages (due to the early lethality limiting later analysis), yet the tendon phenotype of the CKO at corresponding stages is unclear. The stages are not specified and Figure 5 legend is missing. The data shown for the CKO appears to be a much later stage. The authors need to include earlier embryonic and postnatal analysis to demonstrate that stage 1 is normal in these mice and that stage 2 at early postnatal stages is disrupted. It would be valuable to include an analysis of fibronectin at these earlier stages. If over-accumulation of fibronectin is involved in their phenotype, there should be increased fibronectin levels preceding or coincident with the matrix changes. The complete null appears to have disruption in both stages. The authors detect fewer collagen fibrils that are looped and longer in length from embryonic (Figure 2, Figure 2—figure supplement 2, Figure 3, and some of the supplemental figures to Figure 3) and P0 analysis (Figure 1). Postnatal data is given for the *Scx-Cre-Mmp14* conditional knockout, but their tendons appear more significantly affected (comparing Figure 1 with Figure 5). The authors should clarify the function of MMP14 at both embryonic and early postnatal stages, and if their data support a direct or indirect role for MMP14 in these processes.

2) In Figure 1, in the evaluation of the effects of *MMp14* KO on tendon size – is there an effect on the number of cells per tendon?

3) The authors present several mechanisms to account for their phenotype. They mainly propose a model whereby collagen release is mediated by MMP14 possibly through cleavage of a molecular bridge possibly composed of fibronectin or periostin. The authors also show decreased amounts of phospho-Smad2, and comment in the Discussion that MMP14 may function to promote TGF-beta signaling, which then serves to induce tendon fates. Although these are plausible hypotheses, more data are required to support them. A focus on one of the potential mechanisms would help with the clarity of the paper. Both TGF-beta (by phosphor-Smad2 staining) and fibronectin are investigated at 7 weeks (Figure 6), a much later stage than any of the other findings. These differences at later stages make it difficult to distinguish between indirect vs. direct role in causing the phenotype. The authors could investigate fibronectin levels at earlier stages to strengthen their data. Similar analysis of periostin or TGF-beta/phospho-smad2 levels at embryonic and postnatal time points, could be performed in the MMP14 loss of function animals.

4) The tendon phenotype of *Scx-Cre; Mmp14 fl/fl* appears more affected than MMP14 nulls (Figure 1 compared with Figure 5) although the general health of the animal is improved. In Figure 1, the null has larger diameter fibrils than wild type, yet in Figure 5 they are unimodal and smaller. It is unclear if the stages are different as it only states postnatal in the text and the Figure 5 legend is missing.

5) The skeletal phenotype shown in Figure 4 is distracting and does not add to the main focus of the paper. The CKO animals appear smaller than the controls, and the reported skeletal changes could occur in proportion to the decreased animal size. In addition, it is unclear whether the skeletal phenotype is directly related to MMP14 loss in the tendons or due to its loss in other tissues where the *Scx-Cre* transgene may be active. Although this transgene's activity has not been reported, the *Scx-GFP* transgene has been described to be active in the lungs, adrenal glands, and kidneys.

6) The relative weakness of the experiment aimed at identifying collagenous peptides released naturally by MMP-14 by LC-MS/MS is a problem. The authors state that the cleavage may occur at levels that are too low to detect. Mass spectrometry on extracts of connective tissues is difficult; it is possible, that the extraction buffer used (0.1 M Tris buffer pH 7.5) was inappropriate. So the failure to identify the expected peptides does not justify the paragraph title “Cleavage of collagen-I is not required to release collagen fibrils from fibripositors”. In this respect, the data with the *Col-r/r* are much more convincing. As no conclusions can be drawn from the authors' description of the data, this section should be removed from the Results and Discussion.

7) The authors propose that FN cleavage may be key to understanding the phenotype and they have already performed LC-MS/MS to analyze collagen cleavage. It may be useful to determine if in the same analysis they identify FN fragments.

---

## [Author Response]

*1) The authors outline two stages of tendon development: in the first stage (from E14.5-) collagen fibril number is increased, and the second stage, at early postnatal stages, the fibrils are released to the ECM where they grow in length and diameter, and fibripositors disappear. They conclude that MMP14 is required for the progression from stage 1 to stage 2, affecting tissue expansion. However, their data suggest that MMP14 may act at multiple stages of tendon development. There is a lack of clarity on the phenotype and stages analyzed for the* Scx-Cre-Mmp14 *tissue specific knockout compared with the null. Much of their data (*Figures 1, 2 and 3*) are on the phenotype of the* Mmp14*-null mice at embryonic and early postnatal stages (due to the early lethality limiting later analysis), yet the tendon phenotype of the CKO at corresponding stages is unclear. The stages are not specified and*
Figure 5
*legend is missing. The data shown for the CKO appears to be a much later stage. The authors need to include earlier embryonic and postnatal analysis to demonstrate that stage 1 is normal in these mice and that stage 2 at early postnatal stages is disrupted. It would be valuable to include an analysis of fibronectin at these earlier stages. If over-accumulation of fibronectin is involved in their phenotype, there should be increased fibronectin levels preceding or coincident with the matrix changes. The complete null appears to have disruption in both stages. The authors detect fewer collagen fibrils that are looped and longer in length from embryonic (*Figure 2*,*
Figure 2—figure supplement 2*,*
Figure 3*, and some of the supplemental figures to*
Figure 3*) and P0 analysis (*Figure 1*). Postnatal data is given for the* Scx-Cre-Mmp14 *conditional knockout, but their tendons appear more significantly affected (comparing*
Figure 1
*with*
Figure 5*). The authors should clarify the function of MMP14 at both embryonic and early postnatal stages, and if their data support a direct or indirect role for MMP14 in these processes.*

The confusion arose partly because the legend to Figure 5 was missing. We apologise for this oversight. We have also added the developmental stages to all the relevant figure legends.

The absence of the legend to Figure 5 caused confusion about whether or not the tendon-specific *ScxCre;Mmp14*-floxed and global *MMp14* KO embryonic tendons have the same phenotype during embryonic development (i.e. during stage 1). To address this point we have performed new transmission electron microscopy (TEM) on P0 tail tendons from *ScxCre;Mmp14*-floxed and wild type tendon (see Figure 4) and confirmed a similar phenotype with increased fibripositors in *ScxCre;Mmp14*-floxed and global *MMp14* KO (Figure 1).

To address the question regarding fibronectin and periostin accumulation in *ScxCre;Mmp14*-floxed tendons, we have performed new immunofluorescence analyses on *ScxCre;Mmp14*-floxed and wild type littermates from E15.5, P0 and P10 (see Figure 6; Figure 6—figure supplement 1). We could not perform western blot analyses at all of these stages because it is technically difficult to isolate tendons cleanly from embryos. However we have included the western blot analysis for periostin at P7 (Figure 6—figure supplement 1).

*2) In*
Figure 1*, in the evaluation of the effects of* MMp14 *KO on tendon size – is there an effect on the number of cells per tendon?*

We performed further analysis to determine if the cell number in *MMp14* KO tendons at and have incorporated the data into the manuscript (Figure 1—figure supplement 1). The results show no difference in cell number between WT and KO tendons.

*3) The authors present several mechanisms to account for their phenotype. They mainly propose a model whereby collagen release is mediated by MMP14 possibly through cleavage of a molecular bridge possibly composed of fibronectin or periostin. The authors also show decreased amounts of phospho-Smad2, and comment in the Discussion that MMP14 may function to promote TGF-beta signaling, which then serves to induce tendon fates. Although these are plausible hypotheses, more data are required to support them. A focus on one of the potential mechanisms would help with the clarity of the paper. Both TGF-beta (by phosphor-Smad2 staining) and fibronectin are investigated at 7 weeks (*Figure 6*), a much later stage than any of the other findings. These differences at later stages make it difficult to distinguish between indirect vs. direct role in causing the phenotype. The authors could investigate fibronectin levels at earlier stages to strengthen their data. Similar analysis of periostin or TGF-beta/phospho-smad2 levels at embryonic and postnatal time points, could be performed in the MMP14 loss of function animals.*

We agree that the main message of the paper is that MMP14 is essential for the release of collagen fibrils from fibripositors at the end of stage 1, and that the fairly long discussion about TGF-beta is perhaps a side issue that detracts from the paper’s main conclusion. The TGF-beta signalling data have now been removed. We agree that this aspect of the paper was not well supported. However, we have included a short sentence near the end of the Discussion alluding to potential additional factors that might contribute to the tendon phenotype in *Mmp14*-null mice.

The reviewers requested additional data on periostin. We have done this and added new data in Figure 6—figure supplement 1, new sentences in Results and new sentences in Discussion.

*4) The tendon phenotype of* Scx-Cre; Mmp14 fl/fl *appears more affected than MMP14 nulls (*Figure 1
*compared with*
Figure 5*) although the general health of the animal is improved. In*
Figure 1*, the null has larger diameter fibrils than wild type, yet in*
Figure 5
*they are unimodal and smaller. It is unclear if the stages are different as it only states postnatal in the text and the*
Figure 5
*legend is missing.*

The concern that *ScxCre;Mmp*-floxed mice appear to have a more severe phenotype than global *MMp14* KO mice arose largely because the lack of a legend for Figure 5 caused much confusion. We apologise for this omission. *MMp14* KO is lethal at P7; therefore TEM analyses of this mouse was performed mainly at E15.5 (stage 1; most of the data) and P0 (stage 1-stage 2 switch; Figure 1, Figure–figure supplement1, Figure 3—figure supplement 2). *ScxCre;Mmp14*-floxed TEM in Figure 5 was performed at 7 weeks-old. However, we have now rectified these omissions (see points 1 and 2) and performed additional electron microscopy. We now provide better evidence showing that the two mice have the same halted stage 1-stage 2 switch phenotype.

*5) The skeletal phenotype shown in*
Figure 4
*is distracting and does not add to the main focus of the paper. The CKO animals appear smaller than the controls, and the reported skeletal changes could occur in proportion to the decreased animal size. In addition, it is unclear whether the skeletal phenotype is directly related to MMP14 loss in the tendons or due to its loss in other tissues where the* Scx-Cre *transgene may be active. Although this transgene's activity has not been reported, the* Scx-GFP *transgene has been described to be active in the lungs, adrenal glands, and kidneys.*

Although the skeletal data do not form the main focus of the paper, we feel that it is important to report this phenotype. We have now moved these data to Figure 4—figure supplement 2 and added sentences to the Discussion.

*6) The relative weakness of the experiment aimed at identifying collagenous peptides released naturally by MMP-14 by LC-MS/MS is a problem. The authors state that the cleavage may occur at levels that are too low to detect. Mass spectrometry on extracts of connective tissues is difficult; it is possible, that the extraction buffer used (0.1 M Tris buffer pH 7.5) was inappropriate. So the failure to identify the expected peptides does not justify the paragraph title “Cleavage of collagen-I is not required to release collagen fibrils from fibripositors”. In this respect, the data with the* Col-r/r *are much more convincing. As no conclusions can be drawn from the authors' description of the data, this section should be removed from the Results and Discussion.*

We agree that the section “Cleavage of collagen-I is not required to release collagen fibrils from fibripositors” was verbose and weak. We have removed the section. The reviewer(s) did not consider this section contributed to the paper, and we can see their point of view. We agree with their conclusion that the *Col-r/r* data are more convincing.

7) The authors propose that FN cleavage may be key to understanding the phenotype and they have already performed LC-MS/MS to analyze collagen cleavage. It may be useful to determine if in the same analysis they identify FN fragments.

With regards to the mass spectrometry studies on fibronectin, we have now further analysed the data and identified a unique semi-tryptic peptide of fibronectin only present in wild type and not MMP-deficient tendons. We confirmed this finding by analysing fibronectin cleaved by recombinant MMP14 in vitro and have included these data in Figure 6.